# The Hidden Fragility in the Heart of the Athletes: A Review of Genetic Biomarkers

**DOI:** 10.3390/ijms21186682

**Published:** 2020-09-12

**Authors:** Ferdinando Barretta, Bruno Mirra, Emanuele Monda, Martina Caiazza, Barbara Lombardo, Nadia Tinto, Olga Scudiero, Giulia Frisso, Cristina Mazzaccara

**Affiliations:** 1Department of Molecular Medicine and Medical Biotechnology, University of Naples Federico II, 80131 Naples, Italy; barretta@ceinge.unina.it (F.B.); mirrab@ceinge.unina.it (B.M.); barbara.lombardo@unina.it (B.L.); nadia.tinto@unina.it (N.T.); olga.scudiero@unina.it (O.S.); cristina.mazzaccara@unina.it (C.M.); 2CEINGE Advanced Biotechnologies, 80131 Naples, Italy; 3Department of Translational Medical Sciences, University of Campania ‘Luigi Vanvitelli’, Monaldi Hospital, 80131 Naples, Italy; emanuelemonda@me.com (E.M.); martina.caiazza@yahoo.it (M.C.)

**Keywords:** sudden cardiac death, athletes, channelopathies, cardiomyopathies, genetic test, next generation sequencing, preventive medicine

## Abstract

Sudden cardiac death (SCD) is a devastating event which can also affect people in apparent good health, such as young athletes. It is known that intense and continuous exercise along with a genetic background that predisposes a person to the risk of fatal arrhythmias is a trigger for SCD. Therefore, knowledge of the athlete’s genetic conditions underlying the onset of SCD must be extended, in order to develop new effective prevention and/or therapeutic strategies. Arrhythmic features occur across a broad spectrum of cardiac diseases, sometimes presenting with overlapping phenotypes. The genetic basis of arrhythmogenic disorders has been greatly highlighted in the last 30 years, and has shown marked heterogeneity. The advent of next-generation sequencing has constantly updated our understanding of the genetic basis of arrhythmogenic diseases and is laying the foundation for precision medicine. With the exception of a few clinical cases involving a single athlete showing a highly suspected phenotype for the presence of a heart disease, there are few studies to date that analysed the applicability of genetic testing on cohorts of athletes. This evidence shows that genetic testing can contribute to the diagnosis of up to 13% of athletes; however, the presence of clinical markers is essential. This review aims to provide a reference collection on current knowledge of the genetic basis of sudden cardiac death in athletes and to review updated evidence on the effectiveness of genetic testing in early identification of athletes at risk for SCD.

## 1. Introduction

Sudden cardiac death (SCD) is defined, according to the European Society of Cardiology (ESC), as a non-traumatic, unexpected fatal event occurring within 1 h of the onset of symptoms, in an apparently healthy subject. In about half of cases, SCD occurs after symptoms, such as chest pain, wheezing and shortness of breath, racing heartbeat, palpitations, feeling dizzy or fainting. The term “cardiac” can be used if a potentially fatal cardiac condition, congenital or acquired, was identified or the autopsy showed cardiovascular abnormalities. Therefore, an arrhythmic event is the most probably cause of death [1]. Indeed, arrhythmic episodes, particularly ventricular fibrillation (VF), are the main cause of SCD, and the prevention of VF is a challenge for electrophysiologists, since the risk of VF is considered as a surrogate of the risk of SCD [1,2,3].

Since the end of the last century, numerous studies showed that risk of SCD increases when there is family history of this event, indicating that there may be a genetic predisposition for SCD. [1,4,5,6,7]. Furthermore, data in the literature suggested that SCD may occur as a consequence of a trigger factor in the context of electrical or structural heart disorder (genetic or acquired), which acts as a predisposing condition. Known exogenous triggers resulting in the fatal event of SCD may be fever, alcohol intake, electrolyte disorders, hypertensive crisis, infection, cytotoxic drugs, and pregnancy [8,9,10,11,12,13,14,15,16,17,18]. Moreover, regular physical activity is an important exogenous factor affecting the risk of SCD in predisposed subjects. Physical activity reduces cardiovascular mortality, showing favourable effects on several disorders, i.e., obesity, type 2 diabetes, dyslipidaemias, hypertension (American Heart Association recommendations for physical activity in adults and kids. Available online: https://www.heart.org/en/healthy-living/fitness/fitness-basics/aha-recs-for-physical-activity-in-adults, accessed on 26 August 2020). However, in the presence of cardiovascular conditions predisposing to life-threatening ventricular arrhythmias, vigorous physical activity may transiently increase the risk of SCD, both in males and females, as evidenced by Corrado et al. in adolescents and young adults [19].

SCD is the most frequent medical cause of sudden death in athletes, at any age. However, coronary artery disease is the most frequent cause of SCD in athletes over 35 years of age, while genetic disorders represent an important cause of SCD in young athletes [2,19,20].

The main inherited arrhythmogenic conditions predisposing one to SCD can be grouped into two classes: the primary inherited arrhythmia syndromes, the so called ‘channelopathies’ (i.e., long QT syndrome (LQTS), Brugada syndrome (BrS) and catecholaminergic polymorphic ventricular tachycardia (CPVT)), and the cardiomyopathies (i.e., hypertrophic, dilated and arrhythmogenic cardiomyopathy) [21]. Globally, cardiomyopathies and primary inherited arrhythmia syndromes show a prevalence of 3% in the general population, and represent a major cause of cardiac morbidity and mortality in the young athletes. They are classified as Mendelian diseases, generally caused by variations in a single gene: the mutation is present since birth and the risk of disease may be suspected on the basis of family history and inherited pattern [22]. In the last three decades, thanks to the remarkable progress in the field of molecular genetics, molecular bases of inherited arrhythmia syndromes have been revealed, thus encouraging the investigation of DNA biomarkers for diagnostic purposes. The mechanisms responsible for inherited arrhythmic disorders are multiple. Mutations in exonic regions of genes encoding cardiac ion channels or structural proteins have been associated with different inherited cardiac arrhythmias. However, apparently non-inherited forms of cardiac arrhythmias could be associated with changes in the expression levels of transcript for various proteins. Functional polymorphisms may affect gene transcription, RNA processing, post-transcriptional control of gene expression by miRNA, protein synthesis, assembly and post-translational modification and trafficking, producing a arrhythmic substrates [23].

Actually, in the field of inherited arrhythmogenic syndromes, genetic testing can be used to confirm diagnosis in a clinically affected patient, to reach an early and definitive diagnosis in patients showing overlapping or intermediate phenotypes, and to make a differential diagnosis between genetic and non-genetic forms of disease [22]. Furthermore, genetic tests in the cardiovascular field can also be used for predictive purposes. Indeed, molecular studies carried out in the field of hereditary myocardial diseases unveiled that the reduced penetrance and variable expressivity of genotype generate the so-called asymptomatic or pauci-symptomatic carriers, in which the diagnosis may be unknown or delayed but which may still cause risk of sudden death when exposed to trigger factors, such as the practice of strenuous and continuous sports [22,24]. Finally, genetic testing can be applied in relatives of subjects suffering from autopsy-negative SCD or unexplained cardiac arrest, to diagnose a heritable aetiology [25,26], which may be genetically transmissible. This is an important strategy for the early identification of asymptomatic relatives showing a positive test (genotype-positive), who may be predisposed to an increased risk of SCD. Indeed, the European Society of Human Genetics (ESHG) endorsed a guideline to perform post-mortem genetic testing, the so-called “molecular autopsy”, on blood or tissue samples from a suddenly deceased person [27]. Ackerman et al. first suggested that molecular autopsy could determine the aetiology of death in SCD when conventional autopsy resulted negative [28]. Subsequently, data in the literature showed that this approach may identify mutations in cardiomyopathy/channelopathy related genes, in about 30–40% of autopsy-negative sudden death cases, suggesting that an inheritable cardiac disorder was the responsible of SCD in these subjects [29,30,31,32,33,34,35]. Similarly, post-mortem genetic studies identified molecular alterations that can be causative or con-causative of SCD in athletes [36,37,38]. However, in recent years, several studies have addressed the issue of using genetic testing in athletes showing instrumental abnormalities (ECG and/or echocardiogram and/or cardiac magnetic resonance) or family history of cardiomyopathy or SCD, with the aim of early diagnosis of life-threatening cardiac genetic disorders [39,40,41,42,43,44,45,46,47,48,49,50].

On the basis of this evidence, in the review we discuss the genetic basis of SCD in athletes, and the effectiveness of the use of genetic testing in selected populations of athletes.

## 2. Genetic Basis of Inherited Cardiac Channelopathies

Channelopathies are a group of inherited primary electrophysiological disorders that display defects in ion channels and/or their regulatory proteins, without myocardial structural defects. LQTS, BrS, and CPVT are the most frequent cardiac channelopathies identified by genetic testing [51].

### 2.1. Long QT Syndrome (LQTS)

LQTS is a hereditary cardiac electrophysiological disorder characterized by the prolongation of QT interval in an ECG and propensity to tachyarrhythmias, including torsade de pointes, ventricular tachycardia and VF, that might result in SCD. The prevalence of LQTS in the general population has been estimated at 1:2500 [52,53]. LQTS can be suspected on a routine ECG if the corrected QT (QTc) interval is ≥480 ms, and the diagnosis is supported in the presence of additional clinical or ECG criteria, such as T-wave alternans, notched T-wave in at least three leads, low heart rate for age, previous torsade de pointes, syncope, family history of LQTS or SCD [54]. Patients with LQTS may be asymptomatic, or experience palpitations, lipothymia or syncopal episodes. SCD can represent the first manifestation of the disease. Arrhythmic events in LQT1 (*KCNQ1* gene mutation) typically happen in situations of increased adrenergic tone (e.g., exercise, especially swimming), while in LQT2 (*KCNH2* gene mutation) they typically happen as a result of strong emotions (especially sudden auditory stimuli), and in LQT3 (*SCN5A*) during sleep or rest [55]. Mutations associated with the LQTS phenotype have been found in at least 20 genes: *AKAP9*, *ANK2*, *CACNA1C*, *CALM1*, *CALM2*, *CALM3*, *CAV3*, *KCNE1*, *KCNE2*, *KCNH2*, *KCNJ2*, *KCNJ5*, *KCNQ1*, *KCNQ2*, *RYR2*, *SCN1B*, *SCN4B*, *SCN5A*, *SNTA1* and *TRDN* (genes and their function are described in Table 1). To date, about 800 pathogenic variations associated with LQTS have been reported in these genes (ClinVar database, available online: https://www.ncbi.nlm.nih.gov/clinvar/ accessed on 26 August 2020), producing a global diagnostic sensitivity that can reach 85%. However, mutations in the *KCNQ1*, *KCNH2*, *SCN5A* genes, encoding the α subunit of the cardiac potassium channel Kv7.1, α subunit of the cardiac HERG potassium channel and α subunit of the cardiac sodium channel Nav1.5, respectively, are involved in about 75% of LQTS patients [56]. Therefore, the most recent evidence in diagnostic procedure suggested analysing *KCNQ1*, *KCNH2*, *SCN5A* genes only; *CACNA1C*, *CALM1*, *CALM2*, *CALM3*, *KCNE1*, *KCNE2*, *KCNJ2*, *TRDN* genes should be included in syndromic LQTS, or LQTS with atypical features, or in acquired LQTS (i.e., drug- or electrolyte-provoked LQTS) [57]. Of note, homozygous or compound heterozygous frameshift mutations in the *TRDN* gene have been associated to exercise-induced arrhythmias in early childhood [58,59]. Gene mutations in LQTS are typically inherited in an autosomal dominant pattern, except for *KCNE1*, *KCNJ2*, *TRDN* gene mutations, which are inherited in an autosomal recessive manner [57,60]. The proportion of a de novo pathogenic variant is low.

### 2.2. Brugada Syndrome (BrS)

Brugada syndrome is an inherited cardiac ion channel disease caused by mutations in trans-membrane ion channels, leading to an increased risk of cardiac arrhythmias, which may result in SCD. It is characterized by the typical coved-typed ST-segment elevation with a negative T wave in the right precordial leading to ECG without structural cardiac abnormalities [71]. Brugada syndrome shows a prevalence of 0.5 per 1000 in the general population, but it is more prevalent in South-East Asia, where a prevalence of 3.7 per 1000 is reported [72]. Different ECG patterns were described; however, only type 1, with coved ST elevation > 2 mm and negative T wave, is diagnostic for BrS (1). Type 2 (saddleback pattern) with ST elevation > 2 mm and positive T wave or type 3, which is characterized by either a saddleback or coved appearance with an ST elevation < 1 mm, could indicate the disease but requires further confirmation. A combination of the ECG changes and clinical criteria (such as syncope, family history of SCD, etc.) lead to a diagnosis of Brugada syndrome [73]. Actually, asymptomatic patients represent a majority (more than 60%) of newly diagnosed BrS patients (REF), while approximately one third presents with syncope. A minority of patient presents with cardiac arrest or SCD [74], which typically occur during sleep or at rest. More than 25 genes are actually known to be involved in BrS. However, mutations in the *SCN5A* gene are involved in about 20–25% of patients while other genes are rare (<5%) or very rare (<1%), (Table 1). Gene mutations in BrS are inherited with an autosomal dominant pattern; the exception is *KCNE5*-related Brugada syndrome (Locus Xq23), which is inherited in an X-linked manner. The proportion of de novo pathogenic variants is very low (estimated at 1%). Advances in identifying the genes involved in BrS have allowed the use of genetic testing in clinical practice. Molecular genetic testing includes the *SCN5A* exonic coding region; however, a multi-gene NGS panel including genes which encode for sodium, potassium, and calcium channels, or proteins associated with these channels (*ABCC9*, *ANK3*, *CACNA1C*, *CACNA2D1*, *CACNB2*, *FGF12*, *GPD1L*, *HCN4*, *HEY2*, *KCND2*, *KCND3*, *KCNE3*, *KCNE5*, *KCNH2*, *KCNJ8*, *LRRC10*, *PKP2*, *RANGRF*, *SCN1B*, *SCN2B*, *SCN3B*, *SCN5A*, *SCN10A*, *SEMA3A*, *SLMAP*, and *TRPM4*) may be used to identify a pathogenic variant (Table 1) [61,75,76]. About 500 pathogenic variations associated with BrS have been reported in these genes in the ClinVar database (ClinVar database, available online: https://www.ncbi.nlm.nih.gov/clinvar/, accessed on 26 August 2020). The diagnostic sensitivity of the genetic test including all genes, to date, associated to BrS, is nearly 40% [77].

### 2.3. Catecholaminergic Polymorphic Ventricular Tachycardia (CPVT)

CPVT is a primary electrical disease characterized by syncopal episodes occurring during exercise or acute emotion, in individuals without structural cardiac abnormalities. The estimated prevalence of CPVT is 1:5000/1:10,000 people [78]. The clinical manifestations of CPVT typically occur in the first decades of life and are triggered by physical or emotional stress. Patients with CPVT have a normal ECG and echocardiogram; therefore, an exercise stress test that elicits ventricular arrhythmias (bidirectional or polymorphic ventricular tachycardia) is necessary to establish the diagnosis [26]. Mutations in the *RYR2* gene, encoding the Ryanodine receptor 2, are involved in about 60% of patients; mutations in the *CASQ2* gene, encoding the Calsequestrin-2, are involved in about 5% of patients, while other genes (*ANK2*, *CALM1*, *CALM2*, *CALM3*, *KCNJ2*, and *TRDN*) are rare (<5%, Table 1) [79,80]. Therefore, the diagnostic sensitivity of genetic test is slightly higher than 60%, by searching mutations in these genes [81]. *RYR2* gene mutations in CPVT are inherited in an autosomal dominant pattern, while *CASQ2* gene mutations are inherited in an autosomal recessive manner. The proportion of a de novo pathogenic variant is particularly high (around 40%). About 200 pathogenic variations associated with CPVT have been reported in these genes (ClinVar database, available online: https://www.ncbi.nlm.nih.gov/clinvar/, accessed on 26 August 2020).

## 3. Genetic Basis of Inherited Cardiomyopathies

Cardiomyopathies are a group of disorders in which the heart muscle is structurally and functionally abnormal in the absence of other diseases that could cause the observed myocardial abnormality [82]. Cardiomyopathies are classified into five different forms: hypertrophic (HCM), dilated (DCM), restrictive (RCM), arrhythmogenic (ACM) cardiomyopathies, and left ventricular non-compaction (LVNC). Globally, about 90 genes associated with hereditary cardiomyopathies have been identified [83].

### 3.1. Hypertrophic Cardiomyopathy (HCM)

HCM is a common inherited heart disease, affecting at least 1 in 500 people. It is the typical “sarcomeric” cardiomyopathy, most frequently related to mutations in genes encoding sarcomeric proteins, which induce overt structural damage or loss of function of the sarcomere (Figure 1). HCM is defined by the presence of increased left ventricular wall thickness (≥15 mm in adult index cases, or ≥13 mm in relatives of known affected patients, ≥2 standard deviations above the predicted population mean in children) that is not solely explained by abnormal loading condition [84]. Clinical manifestations are highly variable and usually related to the left ventricular outflow tract obstruction, mitral regurgitation, diastolic dysfunction, myocardial ischemia, and arrhythmias, and are represented by dyspnoea, chest pain and syncope [85]. Unfortunately, SCD may be the first manifestation of the disease. There are eight main genes associated, but the most common mutated ones are *MYH7* and *MYBPC3*, encoding the sarcomeric proteins β-myosin heavy chain and cardiac myosin-binding protein C, respectively (Table 1) [86]. More than 450 unique pathogenic variations associated with HCM have been reported in *MYBPC3* and *MYH7* genes (ClinVar database, available online: https://www.ncbi.nlm.nih.gov/clinvar/, accessed on 26 August 2020), while in 10% of HCM cases mutations in *MYL2*, *MYL3*, *TPM1*, *TNNT2*, *TNNI3* and *ACTC1* genes are reported [87] (Table 1). In addition to the eight main genes, other less commonly mutated genes associated with HCM have been identified: *CSRP3*, *FHL1*, *PLN*, *ACTN2*, *CRYAB*, *FLNC*, *MYOZ2*, *MYH6*, *TNNC1*, *TRIM55*, and *TRIM63* (Table 1) [88]. The total diagnostic sensitivity of genetic test is about 65%. HCM has, generally, an autosomal dominant transmission, therefore it affects men and women in the same way; however, clinical diagnosis may be delayed because mutation penetrance is incomplete. In addition, epigenetic influences and environmental factors, such as sports, can influence the development of the pathology [89,90]. About 5% of individuals have compound/digenic heterozygosity [91]. These patients, generally, have a more severe phenotype with earlier onset and faster progression to the end-stage HCM [92].

### 3.2. Dilated Cardiomyopathy (DCM)

DCM is a myocardial disease characterized by ventricular dilation and depressed myocardial performance in the absence of hypertension, congenital, valvular, and ischemic heart disease. Men are more affected than women and DCM represents the most common cardiomyopathy in the world (40/100,000). It represents a heterogeneous disorder, which can be classified into genetic and non-genetic forms. The highest percentage of DCM (50–70%) is covered by genetic factors, while other acquired causes (thyroid pathologies, iron overload, exposure to cardiotoxic drugs, infections) are less frequent. In some cases, it can be asymptomatic, but if untreated it can lead to heart failure with reduced ejection fraction [93] with its classical presentation (dyspnoea, orthopnoea, leg swelling, shortness of breath, etc.). Inherited dilated cardiomyopathy appears to be as a monogenic trait with autosomal dominant, X-linked, autosomal-recessive, or matrilineal pattern of transmission. The molecular basis of DCM is very heterogeneous. To date, at least 50 different genes have been associated with the disease, with the main ones being: *TTN*, *LMNA*, *MYH7*, *TNNT2*, *MYBPC3*, *SCN5A* (Table 1). All these genes show an autosomal dominant inheritance [94]. *TTN* is the most frequently mutated gene with a prevalence of 12–25%, while *LMNA* appears to be the responsible gene in 10% of cases. A disease-causing mutation in *ACTC1*, *MYBPC3*, *MYH7*, *MYL2*, *MYL3*, *TNNI3*, *TNNT2*, and *TPM1* can be identified in 5–10% of cases, and in *SCN5A* in 2–3% of cases. Mutations in the desmosomal genes (*DSC2*, *DSG2*, *DSP*, *JUP*, *PKP2*) are involved in 5% of patients, while other genes are rare (<5%) [95]. Genes coding for cytoskeleton proteins or dystroglycan-associated proteins can be rarely (<1%) mutated in DCM patients [62,95]. More than 200 pathogenic variations associated with DCM have been reported in the ClinVar database (ClinVar database, Available online: https://www.ncbi.nlm.nih.gov/clinvar/, accessed on 26 August 2020), in the two major genes (*TTN*, *LMNA*).

### 3.3. Restrictive Cardiomyopathy (RCM)

Poorly distensible ventricular walls, which oppose diastolic filling, characterize RCM; either a single ventricle (more frequently the left) or both ones can be affected. RCM causes an increase in myocardial stiffness with small increases in ventricular volumes. This complicates the classification of the disease, as the same signs occur in HCM and DCM. The real prevalence of the disease is unknown. The main mutations associated with RCM affect the *MYH7* and *TNNI3* genes [89]. Thirty-seven pathogenic variations associated with RCM have been reported in these genes (ClinVar database, Available online: https://www.ncbi.nlm.nih.gov/clinvar/, accessed on 26 August 2020). A recent study analysed, through NGS technology, 32 independent patients, who fulfilled the diagnostic criteria for idiopathic RCM and demonstrated that causative pathogenic variants can be identified in > 60% of cases in *MYH7*, *DES*, *FLNC*, *MYBPC3*, *LMNA*, *TCAP*, *TNNI3*, *TNNT2*, *TPM1* and *LAMP2* genes. Furthermore, considering the phenotypic overlap of HCM and RCM, the genetic test is fundamental to allow a differential diagnosis between these two pathologies [96].

Cardiac dysfunction may be part of the phenotype of mitochondrial disorders (MD). The most frequent cardiac manifestation of mitochondrial disorders is the HCM, generally associated with mutations in mitochondrial genes *MTCO2*, *MTCO3*, *MTRNR1*, *MTRN2*, *MTTK*, *MTTL1* and nuclear genes *SURF2*, *SCO2*, *MRPL3*, *MRPL44*. More rarely, MD causes dilated cardiomyopathy, restrictive cardiomyopathy, or left ventricular non-compaction [62,97].

### 3.4. Arrhythmogenic Cardiomyopathy (ACM)

Arrhythmogenic cardiomyopathy represents one of the major causes of SCD in the young and athletes [98]. It is a rare disease (prevalence 1:2000–1:5000), generally transmitted in an autosomal dominant manner; however, autosomal recessive inheritance is sometimes reported. ACM is characterized by fibro-fatty replacement of the left and/or right ventricular myocardium, which predisposes patients to heart failure, ventricular arrhythmias and SCD. The main clinical features of ACM are palpitations, syncope, and cardiac arrest in teenagers and adults. Most mutations that cause ACM fall into the desmosomal genes: *PKP2* (30–40%), *DSG2* (3–20%), *DSP* (3–15%), and *DSC2* (1–8%) (Table 1). For all these genes, the transmission is autosomal-dominant, except for *DSC2,* that shows both a dominant and recessive autosomal transmission pattern [95]. As for the desmosomal genes, over 300 unique pathogenetic variants associated with ACM are reported in ClinVar database (ClinVar database, available online: https://www.ncbi.nlm.nih.gov/clinvar/, accessed on 26 August 2020). In addition, mutations in extra-desmosomal genes are reported, such as *TTN*, *FLNC*, *LMNA*, *TMEM43*, *DES*, *PLN*, genes typically associated with DCM, and *SCN5A*, more frequently associated with BrS or LQTS; nevertheless, molecular epidemiology studies showed a low frequency of mutations in these genes. These evidences highlight the genetic overlap of ACM with DCM and channelopathies [99]. Mutations in genes coding proteins involved in other types of cell-cell junctions have been recently described, such as *CADH2* and *CTTNA3*, involved in the formation of adherent junctions, or *TJP1* gene, which encodes zonula occludens-1 intercalated disk protein. However, the prevalence of these ACM form remains to be determined [100,101]. Therefore, the hypothesis that has been postulated is that ACM is not exclusively a desmosomal disease, but it more generally involves the intercalary discs. Interesting, up to 25% of ACM patients shows compound/digenic heterozygosity, which is related to a major risk of life-threatening arrhythmias and SCD [102].

### 3.5. Left Ventricular Non-Compaction (LVNC)

LVNC is a very rare cardiomyopathy, estimated to affect from 8 to 12 per 1 million individuals each year. It is characterized by increased left ventricular trabeculations, a thin normal compacted layer of myocardium, and inter-trabecular recesses communicating with the LV cavity. This cardiomyopathy carries a high risk of malignant arrhythmias, thromboembolic phenomenon and left ventricular dysfunction [103]. The genetic basis of the LVNC remains complex and, similarly to other cardiomyopathies, environmental factors and several modifying genes influence the phenotypic expression [104]. The main genes associated with LVNC are *MYH7*, *MYBPC3*, *ACTC1*, *TNNT2*, *DTNA*, *LDB3*, and *LMNA*, which are also associated with other cardiomyopathies [81]. In most cases, LVNC is transmitted as an autosomal dominant pattern and more than 100 pathogenetic variants in the genes described above are associated with LVNC (ClinVar database, Available online: https://www.ncbi.nlm.nih.gov/clinvar/, accessed on 26 August 2020).

## 4. Diagnostic Yield of Genetic Testing in Athletes

We reviewed the literature data, searching publications regarding the use of genetic tests to diagnose cardiomyopathy/channelopathy in athlete(s) or young subjects practicing sports. The objective was to assess the clinical validity and utility of genetic tests in the diagnostic work-up of athletes referred to tertiary cardiological centres for suspicion of inherited cardiac disease, following the pre-participation screening programme, to early identify high-risk athletes harboring such conditions and provide appropriate advice regarding participation in regular exercise and competitive sport.

### 4.1. Evidence Acquisition: Database Searching, Study Eligibility and Inclusion/Exclusion Criteria

The search followed the Preferred Reporting Items for Systematic Reviews and Meta-Analyses (PRISMA) methodology [105], and covered the PubMed and Scopus databases up to 2020. The search term used was “genetic test/gene testing” combined with “athlete(s)”, “sport”, “cardiomyopathy/cardiomyopathies”, “channelopathy/channelopathies”.

Two authors (N.T. and B.L.) independently chose papers based on the titles and the abstracts, firstly. Then, full texts were analysed. Only English-language publications were considered. We also searched through the bibliographies of selected articles to ensure that no other relevant articles were missed; however, no papers were found by these means. A word file reporting the selected papers was developed and made available to all the co-authors.

To minimize risk of bias and applicability, to be included in the analysis, studies had to report and discuss ethnicity, sex and age of athletes, reasons for enrolment, exhaustive clinical and instrumental evaluation, familiarity with SCD, technology of genetic test. A genetic test was considered positive when it identified actionable variants, i.e., variants classified as pathogenic/likely pathogenic, according to the American College of medical genetics and genomics (ACMG) guidelines [106]. Case reports were included. Studies that did not disclose the clinical/instrumental/genetic investigative modality or presented incomplete data were not included. Papers that reported post-mortem genetic analysis were excluded, as we did not aim to identify the contribution of genetic background in athletes who had unfortunately died suddenly, but we aimed to verify the clinical validity and utility of genetic tests in the setting of athletes’ healthcare if they showed signs/symptoms attributable to a genetic heart disease.

Any issue encountered by an author when extracting the data was discussed collectively and a consensus was adopted to harmonize the extraction process.

### 4.2. Evidence Synthesis

Twelve publications, ranging from 2008 to 2020, were found, meeting the above-mentioned eligibility criteria, in principle. Seven papers depicted eight single case reports of athletes of Caucasian origin [39,40,41,42,43,44]. However, we excluded three among seven, because they did not report technologies used to perform genetic test and/or the specifications of the identified genetic variants [39,40,41]. The remaining are four case reports of white, asymptomatic, amateur athletes, who showed ECG anomalies during the pre-participation screening requiring a thorough cardiac evaluation in tertiary referral centres (Table 2) [24,42,43,44]. Of note, an athlete reported that two relatives died suddenly at a young age [44]. Genetic analyses, performed using Sanger sequencing and/or next-generation sequencing, confirmed the diagnostic suspicion suggested by anamnestic, clinical and instrumental data (i.e., BrS, HCM and ACM, Sick Sinus Syndrome, respectively). Furthermore, the positive genetic test in the athlete made it possible to extend the genetic analysis to relatives, allowing to identify other asymptomatic carriers who had been discouraged from practicing competitive sports.

Five studies carried out a genetic test on athlete populations, including white and black, elite and amateur athletes [45,46,47,48,49,50]. Among these, we excluded the study of Macarie et al. [45], as the authors performed the genetic test using Multiple Ligation Dependent Probe Amplification (MLPA), exclusively, and did not precisely report the mutations found in the athletes. The study of Sheikh et al. did not investigate the SCD familiarity. However, all authors agreed to include it in the review, as the other inclusion criteria are completely met (Table 2).

Kadota et al. conducted, in 2015, the first systematic screening of sarcomere gene mutations in a cohort of Japanese athletes (N: 102) showing ECG abnormalities or mild cardiomegaly in the chest X-ray examination. The selection criteria to enrolment were as described in Pelliccia et al. [107], i.e., ECG patterns that may mimic HCM or ACM, which can be responsible for SCD during physical activity. They found sarcomere gene mutations in five athletes (4.6%), indicating, for the first time, the potential utility of genetic testing in trained athletes to perform a definitive diagnosis of HCM in subjects with early ECG signs of left ventricular wall thickness (Table 2) [46].

Subsequently, Sheikh et al. (2018) analysed 100 consecutive asymptomatic young athletes (50 black and 50 white) showing T-wave inversion (TWI) at ECG and a normal echocardiogram. The comprehensive clinical/instrumental evaluation diagnosed a cardiomyopathy in 21 athletes (19 HCM, 1 LVNC, 1 Fabry disease; 30% among white athletes and 12% among black). Furthermore, using a NGS panel of 311 genes related to cardiomyopathies/channelopathies, a disease-causing variant was identified in 10 athletes: 6 HCM, 1 Fabry disease; 1 LVNC, 1 LQTS and a pathogenic polymorphism in the transthyretin gene, associated with wild-type transthyretin amyloidosis. Of note, the latter two mutations were found in athletes negative at clinical investigation. The diagnostic yield of genetic testing was one half compared to clinical evaluation: however, it contributed to additional diagnoses in 2.5% of athletes with TWI in the absence of a clear clinical phenotype (Table 2) [47].

In 2020, we found two papers related to the issue of clinical significance of genetic screening in athletes (Table 2) [48,50]. Cronin et al. conducted a three-year prospective study, during which they selected 10 white male athletes showing TWI on ECG and phenotypically normal after clinical/instrumental evaluation. Analysing a panel of 133 genes associated with hereditary cardiomyopathies and channelopathies, they identified only variants of uncertain significance (VUS) in half of the athletes; therefore, they concluded that T wave inversion with a normal phenotype might be a benign athletic condition, although studies on larger populations are needed [48]. The recent study of Limongelli et al. assessed the value of genetic testing in the diagnostic flow-chart of athletes [50]. They analysed 61 elite or amateur white athletes, selected due to high suspicion for underlying cardiac disease following the pre-participation screening programme. The selected athletes underwent clinical evaluation and genetic testing, using both Sanger sequencing and NGS panel of 138 genes associated to cardiomyopathies/channelopathies. A total of 23% of the selected population had a definitive diagnosis of inherited cardiomyopathy; of note the diagnostic yield of genetic testing vs. clinical evaluation was 13% vs. 10%. This study supported the use of genetic testing in athletes when a thorough clinical evaluation elicited strong suspicion of cardiac disease but did not lead to a certain diagnosis.

## 5. Discussion

SCD is a catastrophic event, which appears even more devastating when it affects the athlete, who in the collective imagination is the prototype of wellbeing and good health. It is a rare event, accounting for nearly 6% of the total SCDs both in agonist and amateur young athletes [108,109,110]. Primary SCD prevention, identifying athletes who are at increased risk of SCD but who have not had an event, is a very complex issue.

The correlation between the practice of physical activity, especially if strenuous and continuative, and the risk of sudden death, has been known since the end of the last century [19]. Therefore, both the American Heart Association (AHA) and ESC endorsed the pre-participation cardiovascular screening of athletes as a life-saving and cost-effective strategy in athletes, including symptom evaluation, family history, physical examination and, in the ESC guidelines, 12-lead ECG [111,112]. The inclusion of the ECG in the pre-participation screening enhances its sensitivity to the early detection of athletes with a structural or electrical cardiac disease, which commonly are associated with ECG abnormalities. It represents a non-expensive and largely available test, which has proven to be more sensitive compared to history and physical examination protocol [113]. ECG is abnormal in more than 80% of individuals with cardiomyopathies and channelopathies, which are responsible for a large part of SCD in young competitive athletes, and the ECG abnormalities identified may be diagnostic or suspect for inherited cardiac diseases, leading in the latter case to further instrumental investigations, such as echocardiography, cardiac magnetic resonance (CMR), ambulatory ECG monitoring, signal-average ECG, etc. For example, the presence of type I Brugada pattern is diagnostic for BrS or the presence of prolonged QTc in an appropriate clinical context is diagnostic of LQTS. On the contrary, the presence of concomitant inferior and lateral T-wave inversion, ST-segment depression or pathological Q waves in a young athlete is highly suspect, but not diagnostic for HCM [114], and requires further investigations.

Although a comprehensive clinical and instrumental evaluation, based on pre-participation screening and further additional instrumental investigations, can obtain a final diagnosis in a large portion of athletes, in some cases the differential diagnosis between myocardial disease and athlete’s heart is difficult. Therefore, the structural, functional and electrical remodeling of the athlete’s heart may lead to ECG, echocardiographic and CMR abnormalities, which may be difficult to differentiate from an inherited cardiac disease. In this subgroup of patients, genetic testing may be resolutive.

The effectiveness of genetic testing in the pre-participation screening programme has aroused great interest in the world of clinical molecular biology, given the growing evidence that phenotypically silent mutations related to inherited channelopathies and cardiomyopathies can represent a predisposing factor for the appearance of fatal arrhythmias triggered by physical activity [24]. Therefore, a correct and early diagnosis of inherited cardiac disease could reduce the risk of SCD, however, this goal is a real challenge in the world of cardiology and sports medicine, and requires an integrated approach between different disciplines, which increasingly includes knowledge of molecular cardiology.

This review first highlighted the great genetic heterogeneity and complexity that characterizes the molecular basis of inherited channelopathies and cardiomyopathies related to SCD. This complexity imposes the need to refer to experts in the field of molecular cardiology, who can use the most advanced sequencing methods, such as NGS, to sequence multiple genes simultaneously, in the effort to reach the maximum genetic yield [57,61,75,76,88,95,100,101]. Another difficult issue is the interpretation of genetic variants in the context of SCD. Distinguishing the genetic results of clinical utility from the variants of uncertain significance needs expert molecular and biological interpretation, which must be related to detailed clinical data. Sometimes, specific functional analyses may be required [105,115].

Our second aim was to review papers that used genetic testing to identify athletes at risk of developing a potential life-threatening cardiac disease, to verify the applicability of the genetic test in the healthcare of athletes, when the initial pre-participation screening highlights the presence of clear cardiomyopathy, or subtle evidence of clinical markers of inherited disease (including evaluation of family history of hereditary cardiac disorders or SCD).

To date, a limited number of papers have addressed this issue. Several papers reported a single case showing early expression and/or family history of cardiac disease. Genetic analysis, moved by a specific diagnostic suspicion, revealed the presence of a disease-causing variant, and the athletes were disqualified from further competitive sport due to the risk of future occurrence of ventricular arrhythmia.

In the last five years, there has been increasing interest in verifying the roles of genetic testing in the healthcare of athletes at the population level. Some papers applied the genetic test in the diagnostic flow-chart of cohorts of athletes targeted for the evidence of ECG anomalies with/without clinical symptoms. These studies unveiled that the percentage of genotype-positive athletes is significant and contributes as an integrative diagnostic instrument in athletes selected on the basis of a strong clinical suspicion for inherited cardiac disease [39,40,41,42,43,44,45,46,49,50]. Indeed, Cronin et al. demonstrated that, in a small cohort of athletes (N: 10), the evidence of TWI at ECG without a clinical phenotype was not related to a specific genetic background [48]. On the other hand, in the study conducted by Sheikh et al. in 100 athletes with TWI at ECG, the genetic yield was 10%, as 10 athletes revealed a clear disease-causing variant. Of note, eight athletes showed clear cardiac disorders on the basis of comprehensive clinical investigation; two were found mutated in the absence of a clinical phenotype. Therefore, the additional contribution of genetic test in the diagnostic flow-chart of athletes with TWI was 2.5%, in the absence of a clear clinical phenotype [47].

Kadota et al., for the first time, used genetic testing to distinguish hypertrophic cardiomyopathy from the athlete’s heart in 102 Japanese athletes showing features of hypertrophy at ECG or at the chest radiograph [46]. In presence of these specific clinical red flags, the genetic test, performed by sequencing of four sarcomeric genes only, was positive in 4.6% of cases. Finally, 61 athletes with high suspicion of underlying cardiac disease underwent a genetic test in the study of Limongelli et al. [50]. The genetic analyses on this strictly selected population of athletes led to a final genetic diagnosis in eight athletes (8/61, 13%) with clinical suspicion for inherited cardiac diseases. Notably, genetic testing confirmed the presence of cardiac disease in 75% of athletes showing >3 red flags at clinical evaluation. The yield of genetic test was reduced to 14.2% and 8.1% in the presence of two or one red flag(s), respectively. Furthermore, a positive genetic test allows performing family cascade screening, which can diagnose other asymptomatic carrier relatives, thus helping to avoid potential SCD in relatives [50].

In conclusion, the analysed papers, although still few in number, show that it is necessary to correctly select athletes eligible for genetic testing, on the basis of phenotypic feature, to obtain maximum clinical efficacy.

## 6. Conclusions

The identification of specific DNA mutation(s) in athletes showing an overt cardiac disease or a subtle evidence of a cardiac inherited disease or family history of hereditary cardiac disorders/sudden cardiac death is an important step in the integrated flow-chart to reach a definitive diagnosis of channelopathies and cardiomyopathies in these subjects at risk of SCD. This issue represents a real application of predictive medicine in the field of such a dramatic event that could occur during intense physical activity, as is sometimes read about or learned in sports-related clinical practice.

On the other hand, the advent of the NGS has provided an important tool in the molecular genetics of inherited cardiac diseases, providing a relatively inexpensive instrument with which to investigate a broad spectrum of genes involved in cardiomyopathy and ion channel disorders. However, this field is only starting to make substantial inroads.

## Figures and Tables

**Figure 1 ijms-21-06682-f001:**
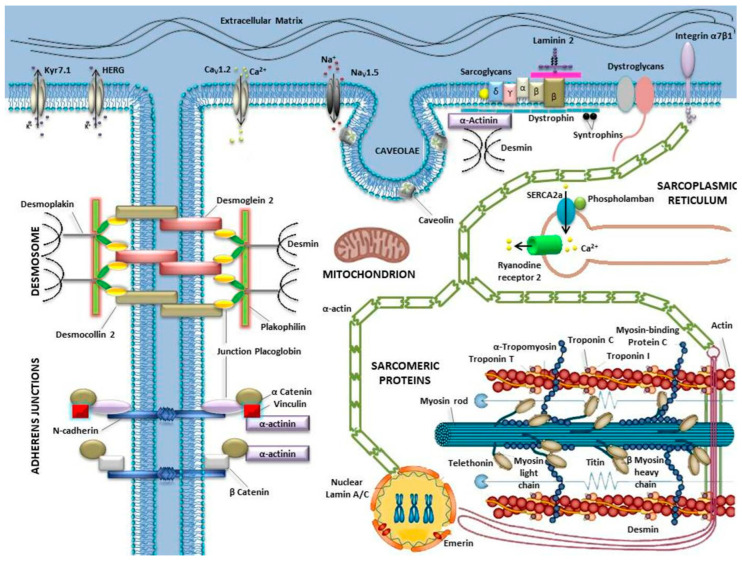
Cartoon of the cardiomyocyte components involved in the molecular mechanisms associated with sudden cardiac death (SCD). The figure illustrates the main ion channels (Kyr7.1, HERG, Cav1.2, Nav1.5, Ryr2) linked to channelopathies (long QT syndrome (LQTS), Brugada syndrome (BrS), catecholaminergic polymorphic ventricular tachycardia (CPVT)); sarcomere proteins associated to hypertrophic cardiomyopathy (HCM); proteins participating in the organization of desmosomes and adherents junctions, linked to arrhythmogenic cardiomyopathy (ACM); the cytoplasmic, mitochondrial, sarcoplasmic reticulum proteins and the proteins of the dystroglycan complex that are associated with the development of dilated cardiomyopathy (DCM).

**Table 1 ijms-21-06682-t001:** Major and Minor Genes Associated to Inherited Channelopathies and Cardiomyopathies.

DISEASE	Gene OMIM ID	GENE	TRANSCRIPT ID	Length bp	PROTEIN	Prevalence	REF.	Inheritance
**Brugada Syndrome (BrS)**	601439	*ABCC9*	ENST00000261200.8	8293	ATP-Binding Cassette, Subfamily C, Member 9	4–5%	[61]	AD
	600465	*ANK3*	ENST00000280772.7	17019	Ankyrin 3 (G)	<5%	[62]	AD
	611875	*CACNA1C*	ENST00000399655.6	13744	α subunit α1C of the Cav1.2 calcium channel	6–7%	[61]	AD
	114204	*CACNA2D1*	ENST00000356860.8	7542	δ subunit Cavα2δ1 of calcium channel	<5%	[61]	AD
	611876	*CACNB2*	ENST00000324631.13	6129	β subunit Cavβ2b of calcium channel	4–5%	[61]	AD
	601513	*FGF12*	ENST00000445105.6	5406	Fibroblast growth factor 12	<5%	[62]	AD
	611777	*GPD1L*	ENST00000282541.10	3947	Glycerol-3-phosphate dehydrogenase 1-like	<5%	[61]	AD
	613123	*HCN4*	ENST00000261917.4	6922	Hyperpolarization-activated cyclic nucleotide-gated channel 4	<5%	[61]	AD
	604674	*HEY2*	ENST00000368364.4	2626	Hairy/Enhancer of Split-related with YRPW motif 2	<5%	[61]	AD
	605410	*KCND2*	ENST00000331113.9	5830	α subunit of the KV4.2 potassium channel	<5%	[62]	AD
	616399	*KCND3*	ENST00000369697.5	7396	α subunit of the KV4.3 potassium channel	<5%	[61]	AD
	613119	*KCNE3*	ENST00000310128.8	3143	β subunit MiRP2 of potassium channel	<5%	[61]	AD
	300328	*KCNE5*	ENST00000372101.3	1473	Potassium channel, voltage-gated, ISK-related family, member 1-like	<5%	[61]	XLD
	152427	*KCNH2*	ENST00000262186.10	4292	α subunit of the HERG potassium channel	<5%	[61] [61]	AD
	600935	*KCNJ8*	ENST00000240662.3	2274	α subunit of the KIR6.1 potassium channel	<5%	[61]	AD
	610846	*LRRC10*	ENST00000361484.5	2344	Leucine-rich repeat-containing protein 10	<5%	[62]	AD
	602861	*PKP2*	ENST00000070846.10	4241	Plakophilin 2	<5%	[61]	AD
	607954	*RANGRF*	ENST00000226105.11	831	RAN guanine nucleotide release factor	<5%	[61]	AD
	612838	*SCN1B*	ENST00000262631.11	1666	β subunit Navβ1 of sodium channel	<5%	[61]	AD
	601327	*SCN2B*	ENST00000278947.6	4937	β subunit Navβ2 of sodium channel	<5%	[61]	AD
	613120	*SCN3B*	ENST00000392770.6	6061	β subunit Navβ3 of sodium channel	<5%	[61]	AD
	601144	*SCN5A*	ENST00000423572.7	8516	α subunit of the Nav1.5 sodium channel	20–25%	[61]	AD
	604427	*SCN10A*	ENST00000449082.3	6626	α subunit of the Nav1.8 sodium channel	2.5–5%	[62]	AD
	603961	*SEMA3A*	ENST00000265362.9	8113	Semaphorin family protein	<5%	[62]	AD
	602701	*SLMAP*	ENST00000659705.1	6240	Sarcolemma-associated protein	<5%	[61]	AD
	606936	*TRPM4*	ENST00000252826.10	4058	Calcium-activated non-selective ion channel	8%	[61]	AD
**Long QT Syndrome (LQT)**	604001	*AKAP9*	ENST00000356239.8	12476	A-kinase anchor protein 9	<5%	[63]	AD
	106410	*ANK2*	ENST00000357077.9	14215	Ankyrin 2 (B)	<5%	[63]	AD
	114205	*CACNA1C*	ENST00000399655.6	13744	α subunit α1C of the Cav1.2 calcium channel	<5%	[63]	AD
	114180	*CALM1*	ENST00000356978.9	4203	Calmodulin 1	<5%	[63]	AD
	114182	*CALM2*	ENST00000272298.12	1210	Calmodulin 2	<5%	[63]	AD
	114183	*CALM3*	ENST00000291295.14	2184	Calmodulin 3	<5%	[62]	AD
	601253	*CAV3*	ENST00000343849.3	1422	Caveolin 3	<5%	[63]	AD
	176261	*KCNE1*	ENST00000399286.3	3505	Potassium voltage-gated channel subfamily E regulatory subunit 1	<5%	[63]	AD/AR
	603796	*KCNE2*	ENST00000290310.	1051	Potassium voltage-gated channel subfamily E regulatory subunit 2	<5%	[63]	AD
	152427	*KCNH2*	ENST00000262186.10	4292	α subunit of the HERG potassium channel	25–30%	[63]	AD
	600681	*KCNJ2*	ENST00000243457.4	5391	Potassium inwardly-rectifying channel, subfamily J, member 2	<5%	[62]	AD/AR
	600734	*KCNJ5*	ENST00000529694.6	6068	Potassium inwardly-rectifying channel, subfamily J, member 5	<5%	[63]	AD
	607542	*KCNQ1*	ENST00000155840.12	3224	Potassium voltage-gated channel subfamily KQT member 1	30–35%	[63]	AD/AR
	602235	*KCNQ2*	ENST00000626839.2	9213	Potassium voltage-gated channel subfamily Q member 2	<5%	[62]	AD
	180902	*RYR2*	ENST00000366574.7	16583	Ryanodine Receptor 2	<5%	[62]	AD
	600235	*SCN1B*	ENST00000262631.11	1666	Sodium voltage-gated channel beta subunit 1	<5%	[62]	AD
	608256	*SCN4B*	ENST00000324727.8	4484	Sodium voltage-gated channel beta subunit 4	<5%	[63]	AD
	600163	*SCN5A*	ENST00000423572.7	8516	α subunit of the Nav1.5 sodium channel	5–10%	[63]	AD
	601017	*SNTA1*	ENST00000217381.3	2211	α1-Syntrophin	<5%	[63]	AD
	603283	*TRDN*	ENST00000334268.9	4627	Triadin	<5%	[62]	AR
**Catecholaminergic Polymorphic Ventricular Tachycardia (CPVT)**	106410	*ANK2*	ENST00000357077.9	14215	Ankyrin 2 (B)	<5%	[64]	AD
	114180	*CALM1*	ENST00000356978.9	4203	Calmodulin 1	<5%	[64]	AD
	114182	*CALM2*	ENST00000272298.12	1210	Calmodulin 2	<5%	[64]	AD
	114183	*CALM3*	ENST00000291295.14	2184	Calmodulin 3	<5%	[62]	AD
	114251	*CASQ2*	ENST00000261448.5	2674	Calsequestrin 2	5%	[64]	AR
	600681	*KCNJ2*	ENST00000243457.4	5391	Potassium inwardly-rectifying channel, subfamily J, member 2	<5%	[64]	AR
	180902	*RYR2*	ENST00000366574.7	16583	Ryanodine Receptor 2	60%	[64]	AD
	603283	*TRDN*	ENST00000334268.9	4627	Triadin	<5%	[64]	AR
**Arrhythmogenic Cardiomyopathy (ACM)**	114020	*CDH2*	ENST00000269141.8	4016	Cadherin 2—N-Cadherin	<5%	[62]	AD
	607667	*CTNNA3*	ENST00000433211.7	10696	Catenin, alpha 3	<5%	[62]	AD
	125660	*DES*	ENST00000373960.4	2243	Desmin	<5%	[62]	AD
	125645	*DSC2*	ENST00000280904.11	12331	Desmocollin 2	1–8%	[62]	AD/AR
	125671	*DSG2*	ENST00000261590.13	5697	Desmoglein 2	3–20%	[62]	AD
	125647	*DSP*	ENST00000379802.8	9697	Desmoplakin	3–15%	[62]	AD
	102565	*FLNC*	ENST00000325888.12	9188	Filamin C	<5%	[62]	AD
	173325	*JUP*	ENST00000393931.8	3505	Junction plakoglobin	<5%	[62]	AD
	150330	*LMNA*	ENST00000368300.9	3178	Lamin A/C	<5%	[62]	AD
	172405	*PLN*	ENST00000357525.6	2989	Phospholamban	<5%	[62]	AR
	602861	*PKP2*	ENST00000070846.10	4241	Plakophilin 2	30–40%	[62]	AD
	180902	*RYR2*	ENST00000366574.7	16583	Ryanodine receptor 2	<5%	[62]	AD
	601144	*SCN5A*	ENST00000423572.7	8516	α subunit of the Nav1.5 sodium channel	<5%	[62]	AD
	190230	*TGFB3*	ENST00000238682.7	2522	Transforming Growth Factor β 3	<5%	[62]	AD
	612048	*TMEM43*	ENST00000306077.5	3230	Transmembrane protein 43	<5%	[62]	AD
	188840	*TTN*	ENST00000589042.5	109224	Titin	0–10%	[62]	AD
**Dilated Cardiomyopathy (DCM)**	601439	*ABCC9*	ENST00000261200.8	8293	ATP-Binding Cassette, Subfamily C, Member 9	<5%	[62]	AD
	602330	*ABLIM1*	ENST00000392955.7	7408	Limatin (actin-binding LIM domain protein)	<5%	[62]	AD
	102540	*ACTC1*	ENST00000290378.6	1382	Actin, alpha, cardiac muscle	<5%	[65,66,67]	AD
	102573	*ACTN2*	ENST00000546208.5	5103	Alpha-actinin 2	<5%	[62]	AD
	606844	*ALMS1*	ENST00000613296.5	12848	ALMS1 centrosome and basal body associated protein	<5%	[62]	AR
	609599	*ANKRD1*	ENST00000371697.3	1979	Ankyrin repeat domain-containing protein 1	<5%	[62]	AD
	608662	*ANO5*	ENST00000324559.9	6742	Anoctamin 5	<5%	[62]	AR
	603883	*BAG3*	ENST00000369085.8	2561	BCL2-associated athanogene	<5%	[62]	AD
	611414	*CALR3*	ENST00000269881.8	1263	Calreticulin 3	<5%	[62]	AD
	114251	*CASQ2*	ENST00000261448.5	2674	Calsequestrin 2	<5%	[62]	AD
	601253	*CAV3*	ENST00000343849.3	1422	Caveolin 3	<5%	[62]	AD
	123590	*CRYAB*	ENST00000533475.6	1117	Alpha B crystallin	<5%	[62]	AD
	600824	*CSRP3*	ENST00000533783.2	1457	Cysteine- and glycine-rich protein 3	<5%	[62]	AD
	600435	*CTF1*	ENST00000279804.2	1664	Cardiotrophin 1	<5%	[62]	AD
	128239	*DAG1*	ENST00000545947.5	5829	Dystroglycan, alpha	<5%	[62]	AR
	125660	*DES*	ENST00000373960.4	2243	Desmin	<5%	[62]	AD
	300377	*DMD*	ENST00000357033.8	13956	Dystrophin	<5%	[62]	XLR
	605377	*DMPK*	ENST00000291270.9	2799	Dystrophia myotonica protein kinase gene	<5%	[62]	AD
	610746	*DOLK*	ENST00000372586.4	2074	Dolichol Kinase	<5%	[62]	AR
	125645	*DSC2*	ENST00000251081.8	12377	Desmocollin 2	<5%	[62]	AD
	125671	*DSG2*	ENST00000261590.13	5697	Desmoglein 2	<5%	[62]	AD
	125647	*DSP*	ENST00000379802.8	9697	Desmoplakin	<5%	[62]	AD
	601239	*DTNA*	ENST00000283365.13	6522	Dystrobrevin, alpha	<5%	[62]	AD/AR
	300384	*EMD*	ENST00000369842.9	1281	Emerin	<5%	[62]	XLR
	603550	*EYA4*	ENST00000355286.12	5701	Eyes absent 4	<5%	[62]	AD
	300163	*FHL1*	ENST00000370683.6	2286	Four-and—a-half LIM domains 1	<5%	[62]	XLD/XLR
	602633	*FHL2*	ENST00000409177.6	4881	Four-and—a-half LIM domains 2	<5%	[62]	AD
	102565	*FLNC*	ENST00000325888.12	9188	Filamin C	<5%	[62]	AD
	614518	*GATAD1*	ENST00000287957.5	4571	GATA Zinc Finger Domain Containing Protein 1	<5%	[62]	AD
	300644	*GLA*	ENST00000218516.4	1318	Galactosidase, alpha	<5%	[62]	XLR
	602366	*ILK*	ENST00000299421.9	1759	Integrin-linked kinase	<5%	[62]	AD
	173325	*JUP*	ENST00000393931.8	3505	Junction plakoglobin	<5%	[62]	AD
	156225	*LAMA2*	ENST00000421865.2	9640	Laminin Alpha, 2	<5%	[62]	AR
	600133	*LAMA4*	ENST00000230538.12	7228	Laminin Alpha, 4	<5%	[62]	AD
	309060	*LAMP2*	ENST00000200639.9	6535	Lysosome-associated membrane protein 2	<5%	[62]	XLR
	605906	*LDB3*	ENST00000429277.6	5436	LIM domain-binding 3	<5%	[62]	AD
	150330	*LMNA*	ENST00000368300.9	3178	Lamin A/C	10%	[62]	AD
	600958	*MYBPC3*	ENST00000545968.6	4217	Myosin-binding protein C, cardiac	<5%	[65,66,67]	AD
	160710	*MYH6*	ENST00000405093.8	5945	Alpha-myosin heavy chain 6	<5%	[62]	AD
	160760	*MYH7*	ENST00000355349.4	6027	Myosin, heavy chain 7, cardiac muscle, beta	<5%	[65,66,67]	AD
	160781	*MYL2*	ENST00000228841.15	783	Myosin light chain 2	<5%	[65,66,67]	AD
	160790	*MYL3*	ENST00000292327.6	885	Myosin light chain 3	<5%	[65,66,67]	AD/AR
	605603	*MYOZ1*	ENST00000359322.5	1300	Myozenin 1	<5%	[62]	AD/AR
	605602	*MYOZ2*	ENST00000307128.6	2549	Myozenin 2	<5%	[62]	AD
	608517	*MYPN*	ENST00000613327.4	6116	Myopalladin	<5%	[62]	AD
	605491	*NEBL*	ENST00000377122.8	9216	Nebulette	<5%	[62]	AD
	613121	*NEXN*	ENST00000330010.12	3195	Nexilin	<5%	[62]	AD
	600584	*NKX2-5*	ENST00000329198.5	1558	NK2 homeobox 5; cardiac specific homeobox 1	<5%	[62]	AD
	605900	*PDLIM1*	ENST00000329399.7	1431	C-terminal LIM domain protein 1	<5%	[62]	AD
	605889	*PDLIM3*	ENST00000284767.12	2802	PDZ and LIM domain protein 3	<5%	[62]	AD
	603422	*PDLIM4*	ENST00000253754.8	2257	PDZ and LIM domain protein 4	<5%	[62]	AD
	602861	*PKP2*	ENST00000070846.10	4241	Plakophilin 2	<5%	[62]	AD
	172405	*PLN*	ENST00000357525.6	2989	Phospholamban	<5%	[62]	AR
	605557	*PRDM16*	ENST00000270722.10	8698	PR domain containing 16	<5%	[62]	AD
	602743	*PRKAG2*	ENST00000287878.9	3279	Protein Kinase, AMP-Activated, Non-Catalytic, Gamma 2	<5%	[62]	AD
	104311	*PSEN1*	ENST00000324501.10	6018	Presenilin 1	<5%	[62]	AD
	600759	*PSEN2*	ENST00000366783.8	2249	Presenilin 2	<5%	[62]	AD
	176876	*PTPN11*	ENST00000351677.7	6073	Protein-Tyrosine Phosphatase, Non-Receptor Type, 11	<5%	[62]	AD
	613171	*RBM20*	ENST00000369519.4	7293	RNA-binding motif protein 20	1–5%	[62]	AD
	609591	*RIT1*	ENST00000368323.8	3390	RIC-like protein without CAAX motif 1	<5%	[62]	AD
	180902	*RYR2*	ENST00000366574.7	16583	Ryanodine Receptor 2	<5%	[62]	AD
	601144	*SCN5A*	ENST00000423572.7	8516	α subunit of the Nav1.5 sodium channel	<5%	[62]	AD
	601411	*SGCD*	ENST00000435422.7	9755	Delta-sarcoglycan	<5%	[62]	AD
	603377	*SLC22A5*	ENST00000245407.8	3277	Solute Carrier Family 22 (Organic Cation Transporter), Member 5	<5%	[62]	AR
	601017	*SNTA1*	ENST00000217381.3	2211	α1-Syntrophin	<5%	[62]	AD
	182530	*SOS1*	ENST00000402219.7	8318	SOS Ras/Rac guanine nucleotide exchange factor 1	<5%	[62]	AD
	607723	*SUN1*	ENST00000401592.6	4010	SAD1 and UNC84 domain-containing protein 1	<5%	[62]	AD
	613569	*SUN2*	ENST00000405510.5	4055	SAD1 and UNC84 domain-containing protein 2	<5%	[62]	AD
	608441	*SYNE1*	ENST00000367255.10	27708	Nesprin 1, Synaptic nuclear envelop protein 1	<5%	[62]	AD
	608442	*SYNE2*	ENST00000358025.7	21842	Nesprin 2, Synaptic nuclear envelop protein 2	<5%	[62]	AD
	300394	*TAZ*	ENST00000601016.6	1906	Tafazzin	<5%	[62]	XLR
	601620	*TBX5*	ENST00000405440.7	3733	T-box 5	<5%	[62]	AD
	604488	*TCAP*	ENST00000309889.3	960	Titin-cap; telethonin	<5%	[62]	AD/AR
	603306	*TCF21*	ENST00000367882.5	1280	Transcription factor 21, epicardin	<5%	[62]	AD
	190230	*TGFB3*	ENST00000238682.7	2522	Transforming Growth Factor β 3	<5%	[62]	AD
	612048	*TMEM43*	ENST00000306077.5	3230	Transmembrane protein 43	<5%	[62]	AD
	188380	*TMPO*	ENST00000556029.5	4242	Thymopoietin	<5%	[62]	AD
	191040	*TNNC1*	ENST00000232975.8	687	Cardiac troponin C	<5%	[62]	AD
	191044	*TNNI3*	ENST00000344887.10	843	Cardiac troponin I3	<5%	[65,66,67]	AD
	191045	*TNNT2*	ENST00000656932.1	1165	Cardiac troponin T2	<5%	[65,66,67]	AD
	191010	*TPM1*	ENST00000403994.9	1217	Tropomyosin 1	<5%	[65,66,67]	AD
	188840	*TTN*	ENST00000589042.5	109224	Titin	12–25%	[65,66,67]	AD
	176300	*TTR*	ENST00000237014.8	616	Transthyretin	<5%	[62]	AD
	193065	*VCL*	ENST00000211998.10	5497	Vinculin	<5%	[62]	AD
**Hypertrophic Cardiomyopathy (HCM)**	102540	*ACTC1*	ENST00000290378.6	1382	Actin, alpha, cardiac muscle	<5%	[68]	AD
	102573	*ACTN2*	ENST00000546208.5	5103	Alpha-actinin 2	<5%	[62]	AD
	603883	*BAG3*	ENST00000369085.8	2561	BCL2-associated athanogene	<5%	[62]	AD
	611414	*CALR3*	ENST00000269881.8	1263	Calreticulin 3	<5%	[62]	AD
	601253	*CAV3*	ENST00000343849.3	1422	Caveolin 3	<5%	[62]	AD
	123590	*CRYAB*	ENST00000650687.2	774	Alpha B crystallin	<5%	[62]	AD
	600824	*CSRP3*	ENST00000265968.9	1283	Cysteine- and glycine-rich protein 3	<5%	[62]	AD
	125660	*DES*	ENST00000373960.4	2243	Desmin	<5%	[62]	AD
	300163	*FHL1*	ENST00000370683.6	2286	Four-and—a-half LIM domains 1	<5%	[62]	XLD/XLR
	602633	*FHL2*	ENST00000409177.6	4881	Four-and—a-half LIM domains 2	<5%	[62]	AD
	102565	*FLNC*	ENST00000325888.12	9188	Filamin C	<5%	[62]	AD
	300644	*GLA*	ENST00000218516.4	1318	Galactosidase, alpha	<5%	[62]	XLR
	602366	*ILK*	ENST00000299421.9	1759	Integrin-linked kinase	<5%	[62]	AD
	605267	*JPH2*	ENST00000372980.4	9502	Junctophilin 2	<5%	[62]	AD
	309060	*LAMP2*	ENST00000200639.9	6535	Lysosome-associated membrane protein 2	<5%	[62]	XLR
	605906	*LDB3*	ENST00000429277.6	5436	LIM domain-binding 3	<5%	[62]	AD
	600958	*MYBPC3*	ENST00000545968.6	4217	Myosin-binding protein C, cardiac	30–40%	[62]	AD
	160710	*MYH6*	ENST00000405093.8	5945	Alpha-myosin heavy chain 6	<5%	[62]	AD
	160760	*MYH7*	ENST00000355349.4	6027	Myosin, heavy chain 7, cardiac muscle, beta	20–30%	[62]	AD
	609928	*MYH7B*	ENST00000262873.12	6293	Myosin Heavy Chain 7B	<5%	[62]	AD
	160781	*MYL2*	ENST00000228841.15	783	Myosin light chain 2	<5%	[62]	AD
	160790	*MYL3*	ENST00000292327.6	885	Myosin light chain 3	<5%	[62]	AD/AR
	606566	*MYLK2*	ENST00000375985.5	2799	Myosin light chain kinase 2	<5%	[62]	AD
	605603	*MYOZ1*	ENST00000359322.5	1300	Myozenin 1	<5%	[62]	AD/AR
	605602	*MYOZ2*	ENST00000307128.6	2549	Myozenin 2	<5%	[62]	AD
	608517	*MYPN*	ENST00000613327.4	6116	Myopalladin	<5%	[62]	AD
	613121	*NEXN*	ENST00000330010.12	3195	Nexilin	<5%	[62]	AD
	605900	*PDLIM1*	ENST00000329399.7	1431	C-terminal LIM domain protein 1	<5%	[62]	AD
	605889	*PDLIM3*	ENST00000284767.12	2802	PDZ and LIM domain protein 3	<5%	[62]	AD
	603422	*PDLIM4*	ENST00000253754.8	2257	PDZ and LIM domain protein 4	<5%	[62]	AD
	172405	*PLN*	ENST00000357525.6	2989	Phospholamban	<5%	[62]	AD
	602743	*PRKAG2*	ENST00000287878.9	3279	Protein Kinase, AMP-Activated, Non-Catalytic, Gamma 2	<5%	[62]	AD
	176876	*PTPN11*	ENST00000351677.7	6073	Protein-Tyrosine Phosphatase, Non-Receptor Type, 11	<5%	[62]	AD
	604488	*TCAP*	ENST00000309889.3	960	Titin-cap; telethonin	<5%	[62]	AD/AR
	612418	*TMEM70*	ENST00000312184.6	2032	Mitochondrial complex V (ATP synthase) deficiency, nuclear type 2	<5%	[62]	AR
	191040	*TNNC1*	ENST00000232975.8	687	Cardiac troponin C	<5%	[68]	AD
	191044	*TNNI3*	ENST00000344887.10	843	Cardiac troponin I3	2–5%	[68]	AD
	191045	*TNNT2*	ENST00000656932.1	1165	Cardiac troponin T2	10–20%	[68]	AD
	191010	*TPM1*	ENST00000403994.9	1217	Tropomyosin 1	2–5%	[68]	AD
	188840	*TTN*	ENST00000589042.5	109224	Titin	<5%	[62]	AD
	176300	*TTR*	ENST00000237014.8	616	Transthyretin	<5%	[62]	AD
	193065	*VCL*	ENST00000211998.10	5497	Vinculin	<5%	[62]	AD
**Restrictive cardiomyopathy (RCM)**	125660	*DES*	ENST00000373960.4	2243	Desmin	<5%	[69]	AD
	102565	*FLNC*	ENST00000325888.12	9188	Filamin C	<5%	[69]	AD
	309060	*LAMP2*	ENST00000200639.9	6535	Lysosome-associated membrane protein 2	<5%	[69]	XLR
	150330	*LMNA*	ENST00000368300.9	3178	Lamin A/C	<5%	[69]	AD
	600958	*MYBPC3*	ENST00000545968.6	4217	Myosin-binding protein C, cardiac	<5%	[69]	AD
	160760	*MYH7*	ENST00000355349.4	6027	Myosin, heavy chain 7, cardiac muscle, beta	<5%	[69]	AD
	604488	*TCAP*	ENST00000309889.3	960	Titin-cap; telethonin	<5%	[69]	AD/AR
	191044	*TNNI3*	ENST00000344887.10	843	Cardiac troponin I3	<5%	[69]	AD
	191045	*TNNT2*	ENST00000656932.1	1165	Cardiac troponin T2	<5%	[69]	AD
	191010	*TPM1*	ENST00000403994.9	1217	Tropomyosin 1	<5%	[69]	AD
**Left ventricular non-compaction (LVNC)**	102540	*ACTC1*	ENST00000290378.6	1382	Actin, alpha, cardiac muscle	<5%	[70]	AD
	601239	*DTNA*	ENST00000283365.13	6522	Dystrobrevin, alpha	<5%	[70]	AD/AR
	605906	*LDB3*	ENST00000429277.6	5436	LIM domain-binding 3	<5%	[70]	AD
	150330	*LMNA*	ENST00000368300.9	3178	Lamin A/C	<5%	[70]	AD
	600958	*MYBPC3*	ENST00000545968.6	4217	Myosin-binding protein C, cardiac	8%	[70]	AD
	160760	*MYH7*	ENST00000355349.4	6027	Myosin, heavy chain 7, cardiac muscle, beta	13%	[70]	AD
	191045	*TNNT2*	ENST00000656932.1	1165	Cardiac troponin T2	<5%	[70]	AD

AD, Autosomic Dominant; AR, Autosomic Recessive; XLR, X-Linked Recessive; XLD, X-Linked Dominant.

**Table 2 ijms-21-06682-t002:** Summary of the Literature Data Concerning the Use of Genetic Tests to Diagnose Cardiomyopathy/Channelopathy in Athlete(s).

Authors	Athletes (N)	Sex	Ethnicity	Age Range (y)	Age (y) or Mean Age (y + SD)	Reason for Enrolment after Clinical Evaluation	SCD Familiarity	Technology of Genetic Test	Positive Genetic Test (%)	Reference
N. Detta et al., 2014	1	1 M	White	NA	12	ECG: HR 50 bpm, with phases of sinus atrial blocks	No	Sanger (*SCN5A, LMNA A/C,EMD, GJ5A, HCN4*)	100	[24]
V. D’Argenio et al., 2018	1	1 M	White	NA	8	ECG: BrS pattern	No	NGS Panel (75 genes)	100	[42]
C. Mazzaccara et al., 2018	1	1 M	White	NA	47	HCM	No	Sanger (8 sarcomeric genes)	100	[43]
G. Limongelli et al., 2020	1	1 M	White	NA	12	Putative ACM	Yes	Sanger (*PKP2, DSP, DSG2, DSC2, RYR2, JUP*). NGS panel (138 genes)	100	[44]
C. Kadota et al., 2015	102	NR	NR	18–28	NR	ECG abnormalities; ECO abnormalities (N = 7)	No	Sanger: *MYBPC3, TNNT2, TNNI3, MYH7* genes	4.9	[46]
N. Sheikh et al., 2018	100	94 M 6 F	50 Black 50 White	14–35	22.7 + 7.3 (Black) 25.1 + 7.1 (White)	ECG: T-wave inversion.	NR	NGS panel (311 genes)	Black: 6 White: 14	[47]
H. Cronin et al., 2020	10	10 M	White	18–54	39	ECG: deep T-wave inversion inferolaterally. All athletes were phenotypically normal	No	NGS panel (133 genes)	0	[48]
G. Limongelli et al., 2020	61	56 M 5 F	White	NR	26 + 12.8	Clinical suspicion for inherited cardiac disease	Yes	NGS panel (138 genes)	13	[50]

Y, Year; SD, Standard Deviation; SCD, Sudden Cardiac Death; NA: Not Applicable; ECG, Electrocardiogram; HR: Heart Rate; bpm: beats per minute; BrS, Brugada Syndrome; NGS: Next Generation Sequencing; HCM: Hypertrophic Cardiomyopathy; ACM, Arrhthmogenic Cardiomyopathy; NR, Not Reported; ECO: Echocardiogram.

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
