# Peer review of "The Hidden Fragility in the Heart of the Athletes: A Review of Genetic Biomarkers"

_ijms, 2020, doi:10.3390/ijms21186682_

Round 1
Reviewer 1 Report
This review focuses on genetic mutations associated with sudden cardiac death (SCD). These genes are described with the hope that genetic profiles for the various types of SCD will be collected and used to predict SCD in young athletes with a familial predisposition for the disease. This is an interesting review about an important topic. There are a couple general comments, one being the value of genetic profiling vs using ECG, especially if there is a change in the ECG rhythm that could be accounted for a change in the expression of or mutation in an ion channel in cardiac tissue. If the authors could add some additional discussion regarding this, it would be helpful. In addition, a great deal of editing needs to be done.
General comments:
- for some of the channelopathies or cardiomyopathies, the authors describe the symptoms of the symptoms, but for others they don’t. Could the authors provide a description for all the disorders described?
- The authors talk about the importance of knowing the genetic profile of young athletes to determine their risk of SCD. However, the disorders associated with mutations in the DNA, or changes in the level of transcription IJMS or ion channels or the proteins necessary for their functioning, are most like associated with changes in ECG, and therefore, this may be an easier, less invasive and more definitive test showing dysfunction
Abstract
Line 17: If the authors can add additional text it would be clearer to say “that predispose a person to the risk…”
Line 19: the authors might consider editing to say “in athletes.” or in ” populations of athletes”
Line 20: It may be clearer to say “we reviewed the data in the literature examining…”
Line 21 and 22: Instead of saying “which can only be overcome by using..” it may be better to say “can only be characterized or described” The way the authors have written that sentence now makes it sound as if this sequencing directly provides a cure. It may provide evidence that can be used to diagnose and prevent sudden cardiac death. But at this point identification of biomarkers is still being completed.
Introduction
Line 29: please define “ESC”
Line 30: after the 1st sentence. The authors may want to list or describe the symptoms that occur in a person have a cardiac event that leads to SCD
Line 31: instead of saying known use identified or diagnosed.
Line 32: The authors can delete “no clear extra cardiac causes” and start a new sentence with “Therefore
Lines 33 and 34: The authors may want to say “episodes” and “are” also define the
Lines 36 – 38 : To shorter this sentence and make it easier to understand, the authors may want to consider editing it to say “ the risk of SCD increases when there is a family history of the disorder, indicating that there may be a genetic predisposition for SCD.
Line 38: the authors may want to consider saying “data in the literature”
Line 47 – 50. Is the risk for SCD increased after vigorous activity in young people with a genetic predisposition, or in all young people?
Line 48: please delete”firstly”
Lines 52 – 53: The authors may want to consider moving this sentence to line 50 where they are talking about children.
Line 53: The authors may want to start a new paragraph when describing the 2 types of cardiac arrhythmias.
Line 61: delete “the”
Line 64 and 65: “for diagnostic purposes”
Line 70: “genetically transmissible”
Lines 54 – 84: The authors talk about genetic mutations that may underlie the development or predisposition to sudden cardiac disease. However, they say that only 50% young people with sudden cardiac disease have a family history or genetic predisposition. The authors may want to add a few sentences saying that non-inherited forms of SCD could be associated with changes in the expression of transcript for various ion channels and other makers. Therefore, know the which DNA show alterations is important, but so is know the level of transcript expression for either normal or altered genes involved in regulating cardiac function.
Line 85”this evidence”
Line 97 – 108 there are a lot of acronyms for the different genes. Please mention that the genes and their function are described in Table 1.
Line 136 : delete totally
Line 160: there are eight main genes associated with…
Line 299: Delete the word “The” and abbreviate SCD
Line 317-381 the authors may want to say “therefore, a correct and early diagnosis of cardiac disease may reduce the risk of SCD…”
Line 321 “pre-participation”
Author Response
Title : ijms-879997
Journal : Int J Mol Sci (MDPI)
Manuscript ID: ijms-879997
Reviewer #1
This review focuses on genetic mutations associated with sudden cardiac death (SCD). These genes are described with the hope that genetic profiles for the various types of SCD will be collected and used to predict SCD in young athletes with a familial predisposition for the disease. This is an interesting review about an important topic. There are a couple general comments, one being the value of genetic profiling vs using ECG, especially if there is a change in the ECG rhythm that could be accounted for a change in the expression of or mutation in an ion channel in cardiac tissue. If the authors could add some additional discussion regarding this, it would be helpful. In addition, a great deal of editing needs to be done.
General comments:
- for some of the channelopathies or cardiomyopathies, the authors describe the symptoms of the symptoms, but for others they don’t. Could the authors provide a description for all the disorders described?
We have added the symptoms and clinical features of all channelopathies and cardiomyopathies described in the text.
LQTS: lines 131-142; The LQTS is a hereditary cardiac electrophysiological disorder characterized by the prolongation of QT interval in an ECG and propensity to tachyarrhythmias, including torsade de pointes, ventricular tachycardia and VF, that might result in SCD. The prevalence of LQTS in the general population has been estimated at 1:2,500 [52, 53]. LQTS can be suspected on a routine ECG if the corrected QT (QTc) interval is ≥480ms, and the diagnosis is supported in the presence of additional clinical or ECG criteria, such as T-wave alternans, notched T-wave in at least three leads, low heart rate for age, previous torsade de pointes, syncope, family history of LQTS or SCD [54]. Patients with LQTS may be asymptomatic, or experienced palpitations, lipothymia or syncopal episodes. SCD can represent the first manifestation of the disease. Arrhythmic event in LQT1 (KCNQ1 gene mutation) typically happen in situation of increased adrenergic tone (e.g., exercise, especially swimming), while in LQT2 (KCNH2 gene mutation) in the contest of strong emotions (especially sudden auditory stimuli) and in LQT3 (SCN5A) during sleep or rest
BrS: lines 174-187; Brugada syndrome is an inherited cardiac ion channel disease caused by mutations in trans-membrane ion channels, leading to an increased risk of cardiac arrhythmias, which may result in SCD. It is characterized by the typical coved-typed ST-segment elevation with a negative T wave in the right precordial leads on ECG without structural cardiac abnormalities [69]. Brugada syndrome shows a prevalence of 0.5 per 1,000 in the general population, but it is more prevalent in South-East Asia where is reported 3.7 per 1,000 [70]. Different ECG patterns were described; however, only type 1, with coved ST elevation >2 mm and negative T wave, is diagnostic for BrS (1). Type 2 (saddleback pattern) with ST elevation >2 mm and positive T wave or type 3, which is characterized by either a saddleback or coved appearance with an ST elevation <1 mm, could indicate the disease but requires further confirmation. A combination of the ECG changes and clinical criteria (such as syncope, family history of SCD, etc.) lead to a diagnosis of Brugada syndrome [71]. Actually, asymptomatic patients represent a majority (more than 60%) of newly diagnosed BrS patients (REF), while approximately one third presents with syncope. A minority of patient presents with cardiac arrest or SCD [72], which typically occur during sleep or at rest.
CPVT: lines 206-212; CPVT is a primary electrical disease characterized by syncopal episodes occurring during exercise or acute emotion, in individuals without structural cardiac abnormalities. The estimated prevalence of CPVT is 1:5,000/1:10,000 people [77]. The clinical manifestations of CPVT typically occur in the first decades of life and are triggered by physical or emotional stress. Patients with CPVT have a normal ECG and echocardiogram; therefore an exercise stress test that elicits ventricular arrhythmias (bidirectional or polymorphic ventricular tachycardia) is necessary to establish the diagnosis.
HCM: lines 231-240; HCM is a common inherited heart disease, affecting at least 1 in 500 people. It is the typical “sarcomeric” cardiomyopathy, most frequently related to mutations in genes encoding sarcomeric proteins, which induce overt structural damage or loss of function of the sarcomere (Fig.1). HCM is defined by the presence of increased left ventricular wall thickness (≥15 mm in adult index cases, or ≥13 mm in relatives of known affected patients, ≥2 standard deviations above the predicted population mean in children) that is not solely explained by abnormal loading condition [83]. Clinical manifestations are highly variable and usually related to the left ventricular outflow tract obstruction, mitral regurgitation, diastolic dysfunction, myocardial ischemia, and arrhythmias, and are represented by dyspnoea, chest pain and syncope [84]. Unfortunately, SCD may be the first manifestation of the disease.
DCM: lines 269-277; DCM is a myocardial disease characterized by ventricular dilation and depressed myocardial performance in absence of hypertension, congenital, valvular, and ischemic heart disease. Men are more affected than women and DCM represents the most common cardiomyopathy in the world (40/100,000). It represents a heterogeneous disorder, which can be classified in genetic and non-genetic forms. The highest percentage of DCM (50-70%) is covered by genetic factors, while other acquired causes (thyroid pathologies, iron overload, exposure to cardiotoxic drugs, infections) are less frequent. In some cases, it can be asymptomatic, but if untreated it can lead to heart failure with reduced ejection fraction [92] with its classical presentation (e.g., dyspnoea, orthopnoea, leg swelling, shortness of breath, etc.).
- The authors talk about the importance of knowing the genetic profile of young athletes to determine their risk of SCD. However, the disorders associated with mutations in the DNA, or changes in the level of transcription IJMS or ion channels or the proteins necessary for their functioning, are most like associated with changes in ECG, and therefore, this may be an easier, less invasive and more definitive test showing dysfunction
We thank you the Reviewer for his/her comment, which improves the paper. We added in the Discussion a paragraph concerning the value of ECG:
lines 491-503. The inclusion of the ECG in the pre-participation screening enhances its sensitivity for early detection of athletes with a structural or electrical cardiac disease, which commonly are associated with ECG abnormalities. It represents a non-expensive and largely available test, which has proven to be more sensitive compared to history and physical examination protocol [113]. ECG is abnormal in more than 80% of individuals with cardiomyopathies and channelopathies, which are responsible of large part of SCD in young competitive athletes, and the ECG abnormalities identified may be diagnostic or suspect for inherited cardiac diseases, leading in the latter case to further instrumental investigations, such as echocardiography, cardiac magnetic resonance (CMR), ambulatory ECG monitoring, signal-average ECG, etc. For example, the presence of type I Brugada pattern is diagnostic for BrS or the presence of prolonged QTc in an appropriate clinical context is diagnostic of LQTS. On the contrary, the presence of concomitant inferior and lateral T-wave inversion, ST-segment depression or pathological Q waves in a young athlete is highly suspect but not diagnostic for HCM [114], and requires further investigations.
Abstract
Line 17: If the authors can add additional text it would be clearer to say “that predispose a person to the risk…”
We have changed according to the Reviewer suggestion (lines 17-19). It is known that intense and continuous exercise along with a genetic background that predisposes a person to the risk of fatal arrhythmias is a trigger for SCD.
Line 19: the authors might consider editing to say “in athletes.” or in ” populations of athletes”
Line 20: It may be clearer to say “we reviewed the data in the literature examining…”
Line 21 and 22: Instead of saying “which can only be overcome by using..” it may be better to say “can only be characterized or described” The way the authors have written that sentence now makes it sound as if this sequencing directly provides a cure. It may provide evidence that can be used to diagnose and prevent sudden cardiac death. But at this point identification of biomarkers is still being completed.
Reviewer #2 required revisions that made it necessary to replace much of the text of Abstract and of the Discussion. Therefore, we have deleted the original sentences in the revised text.
Introduction
Line 29: please define “ESC”
We have defined the ESC abbrevation. Line 44: according to the European Society of Cardiology (ESC),
Line 30: after the 1st sentence. The authors may want to list or describe the symptoms that occur in a person have a cardiac event that leads to SCD.
We thank the Reviewer for the suggestion. We have described the symptoms that occur in a person having a cardiac event that leads to SCD. Lines 46-47: In about half of cases, SCD occurs after symptoms, such as chest pain, wheezing and shortness of breath, racing heartbeat, palpitations, feeling dizzy or fainting.
Line 31: instead of saying known use identified or diagnosed.
We have replaced “known” by “identified”, according to Reviewer’s suggestion. Line 49
Line 32: The authors can delete “no clear extra cardiac causes” and start a new sentence with “Therefore
We have deleted “no clear extra cardiac causes” and started a new sentence with “Therefore”. Lines 49-50: …..or the autopsy showed a cardiovascular abnomalities. Therefore, an arrhythmic event…..
Lines 33 and 34: The authors may want to say “episodes” and “are” also define the
Made: line 51.
Lines 36 – 38: To shorter this sentence and make it easier to understand, the authors may want to consider editing it to say “ the risk of SCD increases when there is a family history of the disorder, indicating that there may be a genetic predisposition for SCD.
We thank you the Reviewer for the suggestion. We have changed the sentence according to his/her suggestion. Lines 54-55: Since the end of the last century, numerous studies showed that risk of SCD increases when there is family history of this event, indicating that there may be a genetic predisposition for SCD.
Line 38: the authors may want to consider saying “data in the literature”
We have changed the sentence according to Reviewer’s suggestion. Lines 56-57: Furthermore, data in the literature suggested that SCD
Line 47 – 50. Is the risk for SCD increased after vigorous activity in young people with a genetic predisposition, or in all young people?
We apologize for the inaccuracy. We have clarified that the risk for SCD increased after vigorous activity in young people with a genetic predisposition. Lines 66-69: However, in the presence of cardiovascular conditions predisposing to life-threatening ventricular arrhythmias, vigorous physical activity may transiently increase the risk of SCD, both in males and females, as evidenced by Corrado et al. in adolescents and young adults
Line 48: please delete ”firstly”
Made
Lines 52 – 53: The authors may want to consider moving this sentence to line 50 where they are talking about children.
We thank you the Reviewer for the suggestion. We have rewritten the sentence to list the causes of sudden death in athletes. Lines 71-75: SCD is the most frequent medical cause of sudden death in athletes, at any age. However, coronary artery disease is the most frequent cause of SCD in athletes over 35 years of age, while genetic disorders represent an important cause of SCD in young athletes
Line 53: The authors may want to start a new paragraph when describing the 2 types of cardiac arrhythmias.
Made (line 76)
Line 61: delete “the”
Made (line 83)
Line 64 and 65: “for diagnostic purposes”
Made (line 87)
Line 70: “genetically transmissible”
Made (lines 106)
Lines 54 – 84: The authors talk about genetic mutations that may underlie the development or predisposition to sudden cardiac disease. However, they say that only 50% young people with sudden cardiac disease have a family history or genetic predisposition. The authors may want to add a few sentences saying that non-inherited forms of SCD could be associated with changes in the expression of transcript for various ion channels and other makers. Therefore, know the which DNA show alterations is important, but so is know the level of transcript expression for either normal or altered genes involved in regulating cardiac function.
We thank you the Reviewer for the suggestion. We added some sentences to clarify the different mechanisms associated to arrhythmic disorders. Lines 87-94: The mechanisms responsible for inherited arrhythmic disorders are multiple. Mutations in exonic regions of genes encoding cardiac ion channels or structural proteins have been associated with different inherited cardiac arrhythmias. However, apparently non-inherited forms of cardiac arrhythmias could be associated with changes in the expression levels of transcript for various proteins. Functional polymorphisms may affect gene transcription, RNA processing, post-transcriptional control of gene expression by miRNA, protein synthesis, assembly and post-translational modification and trafficking, producing a arrhythmic substrates
Line 85: ”this evidence”
Made (line 121)
Line 97 – 108: there are a lot of acronyms for the different genes. Please mention that the genes and their function are described in Table 1.
We added a sentence clarifying that genes and their function are described in Table 1 (line 145-146) “……and TRDN (genes and their function are described in Table 1)”
Line 136: delete totally
Made (line 203)
Line 160: there are eight main genes associated with…
Made (line 240)
Line 299: Delete the word “The” and abbreviate SCD
Made (line 481)
Line 317-381 the authors may want to say “therefore, a correct and early diagnosis of cardiac disease may reduce the risk of SCD…”
We have corrected according to Reviewer suggestion. Line 530-531: Therefore, a correct and early diagnosis of inherited cardiac disease could reduce the risk of SCD,
Line 321 “pre-participation”
Made (lines 552, 555)
Reviewer 2 Report
Dear authors
The review is very important since it discusses an important and current issue related to athletes and a health concern. Nonetheless, the methodology, presentation of results, and discussion were not adequately carried out. Therefore, I invite you to read my comments carefully and correct the manuscript accordingly.
Best regards

Author Response
Title : ijms-879997
Journal : Int J Mol Sci (MDPI)
Manuscript ID: ijms-879997
Reviewer #2
General comments
The abstract should be rewritten
The aim of the study should be clearly defined as a testable aim and not a description of what has been done (review or summarizing published studies)
English should be checked and corrected. Please correct the use of present tense when reporting published studies e.g in introduction
The methodology used to include or exclude studies should be added as an independent section.
Table 1 and figure 1 should be checked by a more specialized reviewer since it could misconduct the reader.
Since authors stated (lines 253-258) that the search followed PRISMA guidelines, the presentation of the results should follow the same guidelines.
Why papers reporting post-mortem genetic analysis were excluded? In your paper you reported that genetic autopsy is important!!!!
The discussion is poorly written and did not focus on the main findings of the review. It should be rewritten.
The studies included in this review were not evaluated regarding their scientific quality and validity. Few information were naively reported. Hence, I suggest that internal validity of the included studies should be carried out and added in a table using EPHPP or Rob tools.
Overall, there was no an “equilibrium” in the review between theoretical concepts (what we know) and the results of the review itself (data analyzed and conclusions of the included studies) that were not critically discussed. The research questions was not clearly answered.
We thank the Reviewer #2 for his/her comments, which made it possible to greatly improve the paper.
According to Reviewer observations, we have rewritten the Abstract and the Discussion. Therefore we have detailed the paragraph 4 (Diagnostic yield of genetic testing in athletes), and have added two sections:
4.1. Evidence acquisition: database searching, study eligibility and inclusion/exclusion criteria;
4.2. Evidence synthesis.
We have detailed the methodology used to include or exclude studies and provided an extensive discussion of the selected studies.
Papers reported post-mortem genetic analysis were excluded, as we did not aim to identify the contribution of genetic background in athletes who had unfortunately died suddenly, but we aimed to verify the clinical validity and utility of genetic tests in the setting of athletes’ healthcare, if they showed signs/symptoms attributable to a genetic heart disease.
Specific comments
Section. Abstract
Line 16-19. This sentence should be rewritten in a correct style. Avoid using “you ....”
Line 20-24. The aim of the study should be testable and clearly defined in relation to your question. Reviewing and summarizing published studies is not a defined aim but a description of what was done.
There is no answer to the question of this study. This is what is expected by the readers at the end of the abstract. This should be added.
We thank the Reviewer for valuable suggestions. We have rewritten the abstract, according to his/her recommendations
Section. Main text
Line 29. ESC should be defined.
We defined the ESC abbrevation (line 44)
Line 44. The reference should be cited correctly. Please refer to the journal instructions
We apologize for the mistake. We have reported the reference according to the journal instructions (lines 63-66).
Line 52-53. A reference should be added.
We added two references (line 75)
Line 92. A reference should be added.
We added a reference (line 129)
Line 104. The reference should be correctly cited (ClinVar database ncbi.nlm.nih.gov/clinvar/)
We apologize for the mistake. We have reported the reference according to the journal instructions (line 147-148)
Lines 100 and 103. To date is repeated.
and
Lines 105-108. The sentence “However, the most recent .......” should be corrected.
We apologize for the repetition. We modified the sentences to better clarify (lines 142-162)
Mutations associated with the LQTS phenotype have been found in at least 20 genes: AKAP9, ANK2, CACNA1C, CALM1, CALM2, CALM3, CAV3, KCNE1, KCNE2, KCNH2, KCNJ2, KCNJ5, KCNQ1, KCNQ2, RYR2, SCN1B, SCN4B, SCN5A, SNTA1 and TRDN (genes and their function are described in Table 1). To date, about 800 pathogenic variations associated with LQTS have been reported in these genes (ClinVar database, available online: ncbi.nlm.nih.gov/clinvar/ accessed on 26 August 2020), producing a global diagnostic sensitivity that can reach 85%. However, mutations in the KCNQ1, KCNH2, SCN5A genes, encoding the α subunit of the cardiac potassium channel Kv7.1, α subunit of the cardiac HERG potassium channel and α subunit of the cardiac sodium channel Nav1.5, respectively, are involved in about 75% of LQTS patients. Therefore, the most recent evidence suggested analysing for molecular diagnostic purposes only KCNQ1, KCNH2, SCN5A genes; CACNA1C, CALM1, CALM2, CALM3, KCNE1, KCNE2, KCNJ2, TRDN genes should be included in syndromic LQTS, or LQTS with atypical features, or in acquired LQTS (i.e. drug or electrolyte-provoked LQTS).
Line 113. A reference should be added.
We added two references (line 166)
Table 01. References should be added regarding the prevalence.
We added a column reporting references
Line 174. A reference should be added
We added a reference (line 254)
Figure 1 should be improved (text size and resolution). A brief description with details should be added in relation to the topic of the review.
We improved Figure 1 (> 1000 pixel; 600 dpi) and modified the figure legend, according to reviewer suggestion
Figure 1. Cartoon of the cardiomyocyte’s components involved in the molecular mechanisms associated with SCD. The figure illustrates the main ion channels (Kyr7.1, HERG, Cav1.2, Nav1.5, Ryr2), linked to channelopathies (LQTS, BrS, CPVT); sarcomere proteins associated to HCM; proteins participating in the organization of desmosomes and adherents junctions, linked to ACM; the multiple cytoplasmic, mitochondrial, sarcoplasmic reticulum proteins and the proteins of the dystroglican complex that are associated with the development of DCM
Lines 220-223. “ACM is a rare disease, because it has a prevalence of 1:2,000-1:5,000 and it has, generally, an autosomal dominant transmission; however, cases in which the transmission is autosomal recessive are reported.” This sentence should be rewritten in a correct style.
We apologize for the mistake. We have rewritten the sentence (lines 313-315)
Arrhythmogenic cardiomyopathy represents one of the major causes of SCD in the young and athletes [98]. It is a rare disease (prevalence 1:2,000-1:5,000), generally transmitted in an autosomal dominant manner; however, autosomal recessive inheritance is sometimes reported. ACM is characterized by fibro-fatty replacement of the left and/or right ventricular myocardium, which predisposes patients to heart failure, ventricular arrhythmias and SCD.
Line 252. 4. Yield of genetic searching for disease-causing mutations in athletes. This title should be rewritten in a more suitable and expressive style.
Thank for the suggestion. We have now replaced “Yield of genetic searching for disease-causing mutations in athletes” with “Diagnostic yield of genetic testing in athletes” (line 354)
Lines 253-258. It should be removed and added to a new section called methodology.
Since you stated that the search followed PRISMA guidelines, the presentation of the results should follow the same guidelines.
Thank you for the observation. We have now rewritten the paragraph and have added two sections:
4.1. Evidence acquisition: database searching, study eligibility and inclusion/exclusion criteria;
4.2. Evidence synthesis.
According to the Referee observations and to the PRISMA guidelines, we have detailed the process of evidence acquisition and evidence synthesis (lines 354-470).
Line 277-278. “Also the athlete analysed ……… » This sentence should be corrected.
In the new version of the paragraph this sentence has been deleted
Lines 299-302. This sentence should be rewritten in a more suitable and correct style by avoiding using informal words.
and
Lines 303-320. I think this part is repeating what was reported in other sections. The discussion should focus on the data analyzed to answer the research question of the review.
and
Lines 321-334. This part gives some “recommendations”. It is strange for me giving recommendations without discussing the main findings of the review!!!
We thank the Reviewer for valuable suggestions. We have rewritten the Discussion, according to his/her recommendations
Round 2
Reviewer 1 Report
This paper looks at biomarkers that may predict sudden cardiac death in athletes. The authors have addressed the reviewers comments.
Reviewer 2 Report
Dear authors
Many thanks for the efforts made to improve the paper
Best regards